# The challenge of SARS-CoV-2 environmental monitoring in schools using floors and portable HEPA filtration units: Fresh or relic RNA?

**Rogelio Zuniga-Montanez**[1], **David A. Coil**[2], **Jonathan A. Eisen**[2,3,4], **Randi Pechacek**[1], **Roque G. Guerrero**[1], **Minji Kim**[1], **Karen Shapiro**[5], **Heather N. Bischel**[1]*

1 Department of Civil and Environmental Engineering, One Shields Avenue, University of California, Davis, California, United States of America, 2 Genome Center, University of California, Davis, California, United States of America, 3 Department of Medical Microbiology and Immunology, School of Medicine, University of California, Davis, California, United States of America, 4 Department of Evolution and Ecology, University of California, Davis, California, United States of America, 5 Department of Pathology, Microbiology and Immunology, School of Veterinary Medicine, University of California, Davis, California, United States of America

* hbischel@ucdavis.edu

**Data Availability Statement:** All relevant data are within the paper and its Supporting information files.

## Abstract

Testing surfaces in school classrooms for the presence of SARS-CoV-2, the virus that causes COVID-19, can provide public-health information that complements clinical testing. We monitored the presence of SARS-CoV-2 RNA in five schools (96 classrooms) in Davis, California (USA) by collecting weekly surface-swab samples from classroom floors and/or portable high-efficiency particulate air (HEPA) units (n = 2,341 swabs). Twenty-two surfaces tested positive, with qPCR cycle threshold (Ct) values ranging from 36.07–38.01. Intermittent repeated positives in a single room were observed for both floor and HEPA filter samples for up to 52 days, even following regular cleaning and HEPA filter replacement after a positive result. We compared the two environmental sampling strategies by testing one floor and two HEPA filter samples in 57 classrooms at Schools D and E. HEPA filter sampling yielded 3.02% and 0.41% positivity rates per filter sample collected for Schools D and E, respectively, while floor sampling yielded 0.48% and 0% positivity rates. Our results indicate that HEPA filter swabs are more sensitive than floor swabs at detecting SARS-CoV-2 RNA in interior spaces. During the study, all schools were offered weekly free COVID-19 clinical testing through Healthy Davis Together (HDT). HDT also offered on-site clinical testing in Schools D and E, and upticks in testing participation were observed following a confirmed positive environmental sample. However, no confirmed COVID-19 cases were identified among students associated with classrooms yielding positive environmental samples. The positive samples detected in this study appeared to contain relic viral RNA from individuals infected before the monitoring program started and/or RNA transported into classrooms via fomites. High-Ct positive results from environmental swabs detected in the absence of known active infections supports this conclusion. Additional research is needed to

**Funding:** Funding was provided by Healthy Yolo Together/Healthy Davis Together. The funders had no role in study design, data collection and analysis, decision to publish, or preparation of the manuscript.

**Competing interests:** The authors have declared that no competing interests exist.

differentiate between fresh and relic SARS-CoV-2 RNA in environmental samples and to determine what types of results should trigger interventions.

## 1 Introduction

First detected in December 2019, the coronavirus SARS-CoV-2 (the causative agent of COVID-19) has been responsible for the largest pandemic in a century. Early in the pandemic there were broad concerns that surfaces might serve as a source of transmission of SARS-CoV-2. However, while some early studies suggested that SARS-CoV-2 virus could remain infectious on surfaces for days [1–4], there has been no indisputable evidence for surface-to-person transmission. Nevertheless, the stability of SARS-CoV-2 RNA on surfaces suggests that environmental monitoring via surface swabs could support the COVID-19 response.

Screening environmental samples such as wastewater and surface swabs for the presence of SARS-CoV-2 RNA provides indirect evidence of the number of infected people shedding the virus in the vicinity [5–8]. Because reverse transcriptase quantitative polymerase chain reaction (RT-qPCR) tests for SARS-CoV-2 RNA are widely available and relatively inexpensive, environmental monitoring has become a cost-effective complement to clinical testing [4, 9]. Environmental monitoring also avoids issues associated with informed consent, sample collection, operational logistics, and equity that can slow or constrain clinical-testing programs [10, 11].

The value of environmental monitoring has been most clearly demonstrated with wastewater surveillance [12, 13]. But while wastewater surveillance can inform policy and action at regional, city, neighborhood, and building levels, it cannot provide information about virus presence in interior spaces (e.g., building floors and rooms) and can provide only very limited assistance in identifying potential virus exposures. Surface sampling is a type of environmental monitoring that could fill this gap by providing decision-makers information at another level of resolution (between larger-scale environmental monitoring and individual clinical testing).

Environmental sampling for SARS-CoV-2 through high-touch surface testing has been examined by both academic research-based projects [4, 14] as well as via companies offering monitoring services [5]. Several groups, including ours, have also evaluated viral RNA in HVAC systems with mixed results [15–19] or with portable air samplers [20, 21]. Complexities of HVAC systems (e.g., shared air between rooms, timed operation, difficult to access, variable filter types) have prevented filter-based monitoring from being widely deployed. Portable high-efficiency particulate air (HEPA) filtration units have the potential to be an effective mitigation strategy for SARS-CoV-2 transmission indoors [22], and their widespread deployment through the pandemic creates a new opportunity for filter-based environmental monitoring.

We conducted a SARS-CoV-2 environmental monitoring study from January to August 2021 where we systematically collected floor and/or HEPA filter swab samples in five elementary schools. 2,341 swab samples were obtained using nylon fiber oral swabs, and SARS-CoV-2 RNA was quantified through RT-qPCR. We compared the efficacy of floor and HEPA-filter samples for detecting SARS-CoV-2 RNA and COVID-19 cases in two of the schools. We hypothesized that HEPA filter sampling would be a more efficient strategy to detect infected individuals than floor sampling since SARS-CoV-2 virions would concentrate on the external surface of the filters as air circulated through the units.

## 2 Materials and methods

### 2.1 Validation of SARS-CoV-2 detection on surfaces and HEPA air purification units

Prior to beginning the sampling campaign, we validated the detection of SARS-CoV-2 on surfaces using opportunistic sampling in two locations within six days after one or more clinical COVID-19 cases were identified. The sampling was conducted once in a house and twice in a school classroom. Forty-six samples were collected from different surfaces and tested, including a portable HEPA filtration unit (MA-40, Medify Air, USA) that was located in the classroom (S2 Table in S1 File). The HEPA filter was dismantled for testing, collecting samples from the outer grill cover, pre-filter mesh, and H13 HEPA filter surface. Detailed information on the validation sampling can be found in the section S1.1 in S1 File.

### 2.2 Sampling framework and locations

We partnered with five schools in Davis, California, USA, to conduct weekly SARS-CoV-2 environmental monitoring using floor and HEPA filter swab samples from January to August 2021. All schools had in-person teaching and pandemic control plans and policies in place. One of two main sampling strategies was applied at each school; (1) only floor or (2) floor and HEPA filter sampling. The environmental sampling strategy at each school is summarized in Table 1. No personal information was collected on any individuals in the schools. The University of California, Davis IRB Administration determined that the study design was exempt from IRB review and approval.

### 2.3 HEPA filter and surfaces sampling

Environmental samples were collected using nylon fiber oral swabs with an ABS handle (Miraclean Technology Co. Ltd, China) that were pre-moistened in DNA/RNA Shield (Zymo Research, USA) before collecting the samples. For floor samples, a square area of approximately 10 cm x 10 cm in the center of the room was thoroughly sampled while rotating the swab. For other surfaces or items, a similar or smaller area, depending on surface or item size, was swabbed in a similar manner to floor samples.

  Two models of portable air purification units equipped with HEPA filters were used in the classrooms, MA-40 (Medify Air, USA) and AeraMax 300 (Fellowes, USA) (Table 1). The manufacturers report clean air delivery rates (CADR) of 380 $m^3$/h for the Medify MA-40 and 319–333 $m^3$/h for the AeraMax 300. Teachers were responsible for running the air purifiers when the classrooms were occupied. To sample the portable HEPA filters, the intake side filter cover was removed and the whole pre-filter mesh (~823 cm2) was thoroughly sampled by rotating the swab. After sample collection, the swab tip was snapped off into a sample tube containing 500 μl of DNA/RNA Shield by bending and rolling the swab at the 30 mm breakpoint without touching the sample. The swab tip was preserved in the DNA/RNA Shield until laboratory processing. All surfaces sampled were wiped down with 75% ethanol wipes (Zhejiang Youquan Care Products Technology Co., Ltd., China) after sample collection.

  The sampling at Schools A, B, C, D, and E was conducted by either the school or Healthy Davis Together (HDT) personnel and sampling kits were prepared and delivered to each location weekly. We created instructional videos to show the sampling teams how to collect HEPA filter and floor samples (section S1.2 in S1 File).

**Table 1. HEPA filter and floor environmental monitoring strategies for the detection of SARS-CoV-2 in five K-6 and K-8 schools.**

| School (School type) | Preliminary testing period | Weekly sampling period | Concluding sampling episode | Number of classrooms sampled | Other rooms sampled | Approximate Enrollment | Samples collected per room and collection periods | Number of rooms with air purifiers (AP) sampled and AP models |
|---|---|---|---|---|---|---|---|---|
| A (K-6) | - | January 28 to August 12, 2021 | - | 12 | 6 | 135–138 students | • One floor sample per room. January 28 to August 12, 2021 <br> • One HEPA filter sample per room. May 20 to August 12, 2021 | • Rooms: 18 <br> • AP models: 12 MA-40 6 AeraMax 300 |
| B (K-6) | - | January 28 to August 12, 2021 | - | 3 | 7 | 47–48 students | • One floor sample per room. January 28 to August 12, 2021 <br> • One HEPA filter sample per room. May 20 to August 12, 2021 | • Rooms: 7 <br> • AP model: MA-40 |
| C (K-8) | February 23, 2021 (11 rooms)[a] | March 2 to May 25, 2021 | - | 10 | 1 | 259 students | • One floor sample per room. March 2 to May 25, 2021 | • Rooms: 0 |
| D (K-6) | March 10 to March 30, 2021 (2 rooms)[b] | April 13 to June 08, 2021 | June 18, 2021 (8 rooms)[d] | 25 | - | 380 students | • One floor and two distinct HEPA filter samples per room. March 10 to June 18, 2021 | • Rooms: 25 <br> • AP model: MA-40 |
| E (K-6) | March 10 to March 30, 2021 (1 room)[c] | April 13 to June 07, 2021 | - | 32 | - | 450 students | • One floor and two distinct HEPA filter samples per room. March 10 to June 07, 2021 | • Rooms: 32 <br> • AP model: MA-40 |

[a]Preliminary testing conducted in the same rooms as weekly sampling; however, only a single surface per room was swabbed. These surfaces included walls, desks, sink counters, floors, door handles, cabinets, and tables.

[b]Preliminary testing conducted in Rooms 11 and 20.

[c]Preliminary testing conducted in a room not included in the weekly sampling.

[d]Concluding sampling session at the end of the school year was conducted in Rooms 5, 6, 9, 13, 15, 16, 17 and 24. A door jam, teacher desk, center surface (desk or projector), floor, and two air filter samples were collected from each room.

## 2.4 RNA extraction and RT-qPCR for surface and air filter swab samples

Samples were received on the same day as sample collection, stored at room temperature for up to 4 hours and processed. DNA/RNA Shield transport media has been demonstrated to stabilize SARS-CoV-2 RNA at ambient temperatures for up to 28 days [23]. Before RNA extraction, samples containing the swab tip in DNA/RNA Shield were vortexed at a medium-high to high speed for 10 minutes to suspend and homogenize the particles collected. The samples were then centrifuged at 10,000 x g for 1 minute to remove bubbles that formed during vortexing. Samples collected through February 11, 2021 were extracted manually utilizing the Pure-Link Viral RNA/DNA kit (Thermo Fisher Scientific, USA) according to manufacturer instructions and starting with a 200 µl sample volume. The validation test samples collected after known positive exposures, as well as all samples collected on March 30, 2021 were also manually extracted. Samples collected from February 12, 2021 through the end of the study were extracted using the MagMAX Microbiome Ultra Nucleic Acid Isolation Kit (Applied

**Table 2. RT-qPCR primers and probe used for the detection of SARS-CoV-2 and φ6 bacteriophage in environmental samples.**

| Target | Oligo | Oligonucleotide sequence (5'-3') | Final concentration (nM) | Amplicon length (bp) | Reference |
|---|---|---|---|---|---|
| SARS-CoV-2 S gene | Forward primer | CCTACTAAATTAAATGATCTCTGCTTTACT | 400 | 157 | (Chan et al. 2020; Horve et al. 2021) [25] |
| | Reverse primer | CAAGCTATAACGCAGCCTGTA | 400 | | |
| | Probe | FAM-CGCTCCAGGGCAAACTGGAAAG-BHQ1 | 200 | | |
| φ6 bacteriophage | Forward primer | TGGCGGCGGTCAAGAG | 400 | 100 | modified from [26] |
| | Reverse primer | GGATGATTCTCCAGAAGCTGCT | 400 | | |
| | Probe | FAM-GTCGCAGGTCTGACACT-MGB | 80 | | |

Biosystems, USA) and a KingFisher Flex automated purification system (Thermo Fisher Scientific, USA). The MagMAX_Microbiome_Stool_Flex.bdz nucleic acid isolation protocol (Applied Biosystems, USA) was utilized, with minor modifications. In brief, the sample lysis step was not conducted as lysis was achieved through the use of DNA/RNA Shield and vortexing. The sample plate was loaded with 200 μl of sample and 260 μl of Binding Bead Mix. RNA extracts were eluted with 100 μl of Elution Solution and stored at -80 ˚C prior to RT-qPCR. The detailed manual and automated extraction protocols are available in the section S1.3 in S1 File. Negative and positive extraction controls were tested on most Kingfisher plates. The negative extraction control consisted of nuclease-free water and was always negative for SARS-CoV-2 by RT-qPCR. The positive extraction control consisted of a swab sample collected weekly from a MA-40 HEPA air filter continuously operated in the laboratory where all samples were processed. The positive extraction control sample was spiked with φ6 bacteriophage as a spike-recovery control [24], which was positive for φ6 by RT-qPCR in every sample. Table 2 shows the φ6 primers and probes used, with further detail in the S1 File.

Extracts were thawed on ice after removal from -80 ˚C. All extracts were analyzed by RT-qPCR targeting the spike glycoprotein (S) gene of SARS-CoV-2 [25] using the Luna Universal Probe One-Step RT-qPCR Kit (New England Biolabs Inc., USA). Each 20 μl reaction contained 10 μl Luna Universal Probe One-Step Reaction Mix (2X), 1 μl Luna WarmStart RT Enzyme Mix (20X), specified concentrations of 1.5 μl combined primer/probe mix (Table 2), 2.5 μl nuclease-free water, and 5 μl RNA extract. Extractswere analyzed in triplicate. Duplicates of positive controls (SARS-CoV-2 RNA extract generously provided by the University of Oregon) and no template controls (nuclease-free water) were included with each qPCR plate. The RT-qPCR assays were performed using aStepOnePlus Real-time PCR System (Applied Biosystems, USA). The thermal cycling conditions were 55 ˚C for 15 minutes and 95 ˚C for 2 minutes, followed by 40 cycles at 95 ˚C for 10 seconds and 60 ˚C for 60 seconds. Samples with at least one of three technical replicates with a cycle threshold (Ct) value lower than 40 were considered positive for SARS-CoV-2.

## 2.5 Clinical testing and reporting of COVID-19 positive individuals by the schools

Schools A, B, C, D, and E were offered weekly free COVID-19 clinical testing through HDT [27, 28]. HDT offered additional on-site clinical testing at Schools D and E. Data on positive COVID-19 cases and on-site clinical testing participation were provided by administrators at each school or from the school district.

## 3 Results and discussion

### 3.1 Validation of environmental detection of SARS-CoV-2 RNA after known exposures

Detection of SARS-CoV-2 RNA on environmental surfaces using RT-qPCR has been demonstrated for a wide range of contexts and surface types [14, 16, 18, 29]. We validated our sampling and analytical protocol by sampling in two locations (a private residence and a school classroom) where at least one COVID-19 positive individual was known to have been present within the past week. First, we sampled surfaces at a house where an asymptomatic COVID-19 positive person had been previously present for 2 hours. Of the ten surface swab samples collected four days after the exposure, the underside of the chair where the COVID-19 positive person had sat and the floor underneath this space were positive for the virus (S2 Table in S1 File), with Ct values of 37.2 and 37.6, respectively. Our results confirmed that SARS-CoV-2 RNA could be detected on surfaces a few days after the exposure, including surfaces that are not frequently touched (e.g., the floor). High-frequency touched surfaces in workplace environments have tested positive for the SARS-CoV-2 virus days after the detection of a positive individual [5]. Positive samples have also been collected from no-touch surfaces up to 27 days after an individual was diagnosed with COVID-19 [30].

Second, we sampled a classroom where two students tested positive after attending school. We conducted two sampling episodes, two and six days after the last day of student attendance. In both sampling episodes, the undersides of the chair, desk and tool box of one of the COVID-19 positive students were positive for SARS-CoV-2 (S2 Table in S1 File), with Ct values of 35.6, 36.8, and 37.6 during the first episode, and 36.2, 36.5, and 36.8 during the second episode, respectively. A portable HEPA filter that was operating in the room during and prior to the exposure was dismantled, swabbed, and tested for SARS-CoV-2. Three samples were collected from each of the outer grill cover and pre-filter mesh, and four samples from the H13 HEPA filter. One, three and two of those samples, respectively, were positive for SARS-CoV-2 (S2 Table in S1 File), with Ct values ranging from 36.1 to 38.7. Twenty other samples collected in the classroom during the two sampling episodes tested negative for SARS-CoV-2 (S2 Table in S1 File).

These preliminary results provided further evidence for the stability of SARS-CoV-2 RNA in the environment at least six days after deposition and validated the potential use of portable HEPA filters for environmental monitoring of COVID-19. Environmental monitoring for the presence of SARS-CoV-2 RNA through air sampling has been demonstrated in clinical and transportation settings using diverse air samplers [21, 31, 32], and by swabbing or vacuuming HVAC systems [15, 32]. However, acquiring and deploying air samplers is challenging due to high costs and noise levels of these instruments. It's possible to mitigate the noise problem by situating air samplers inside the HVAC system as in [21]. However, HVAC systems can be difficult to access, interpreting results can be challenging due to shared airflow among different rooms, and distance between the sampling location and infected individuals influences detection. Deploying air purifiers equipped with HEPA filters into rooms themselves is a lower-cost, more accessible, quieter, and better-targeted alternative to using air samplers or HVAC systems for environmental monitoring. Deploying HEPA filters for environmental monitoring has the added benefit of reducing risk of airborne SARS-CoV-2 transmission [22, 33].

### 3.2 Environmental monitoring for SARS-CoV-2 in K-6 and K-8 schools using floor swab samples

We partnered with five schools to implement surface-based SARS-CoV-2 environmental monitoring. In School A (n = 470 swabs), floors were monitored in eighteen rooms weekly from

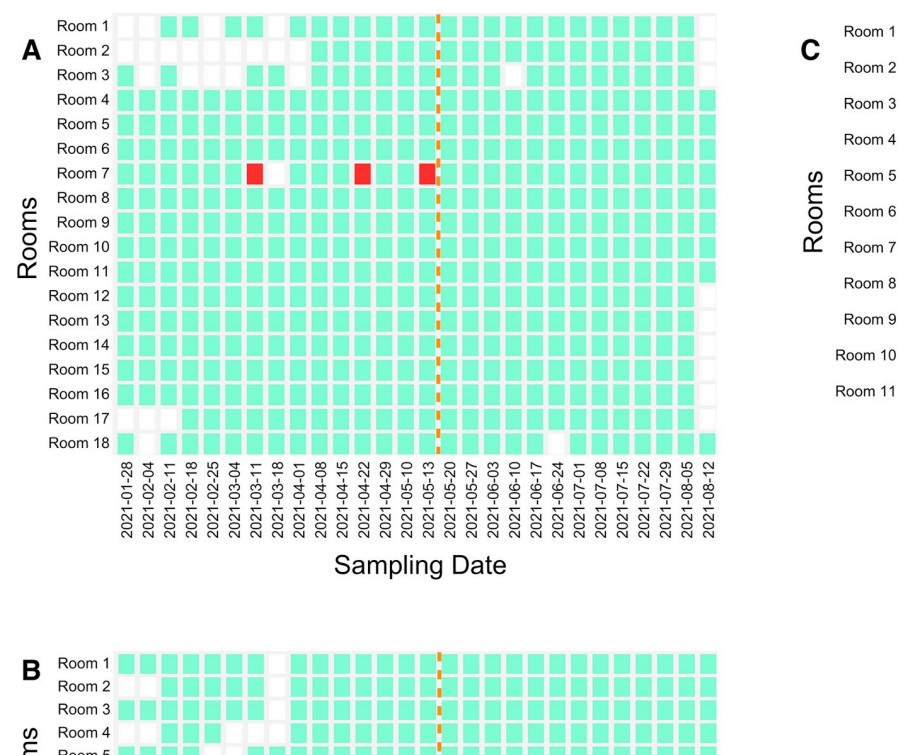

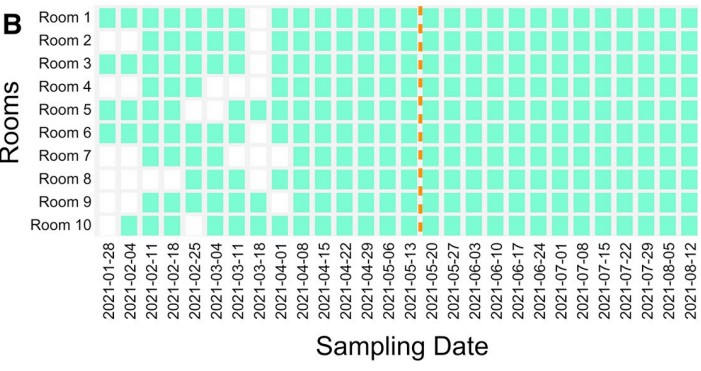

**Fig 1.** Positive and negative rooms for SARS-CoV-2 based on floor samples collected in (**A**) School A, (**B**) School B, and (**C**) School C throughout the environmental monitoring study. Episodes with a positive floor sample are marked in red, negative episodes in green, and episodes where no sample was collected are in white. Air filter sampling in Schools A and B started on May 5, 2021 and is denoted by the orange line. No positive air filter samples were detected.

January 28 to August 12, 2021. SARS-CoV-2 positive environmental samples were intermittently detected only in Room 7 on March 11, April 22, and May 13, 2021 with negative results in between these dates (Fig 1A). Two of 18 floor samples collected on May 6, 2021 in School A were also positive, but the tube labels were unidentifiable (excluded from Fig 1A). A fresh set of samples collected from the 18 rooms on May 10 were negative. No clinical positive cases were reported from School A classrooms when surface samples were positive. However, an individual from Room 7 was confirmed to be positive for COVID-19 on February 3, 2021 (personal communication with school administrator), close to a month before the first positive environmental detection on March 11, 2021.

It is possible that relic RNA shed weeks before the initial detection led to intermittent positive tests during our study. Relic SARS-CoV-2 RNA—RNA in the environment from degraded and non-infectious virus that has little significance to public health—has been detected from a few weeks after the recovery or termination of quarantine for patients [34, 35] and up to two months after symptom onset in a single household with two isolated patients [15]. All

identifiable positive environmental samples in School A originated from a single room and showed intermittent positive results throughout a period of more than two months. The result would not be considered a "false positive" in the technical sense of the terms since we believe it represents the detection of RNA. Air filter sampling began in parallel to the floor sampling in School A on May 20, 2021. All HEPA filter and floor samples collected between May 20 and August 12, 2021 were negative for SARS-CoV-2.

In School B (n = 252 swabs), ten rooms were monitored from January 28 to August 12, 2021 through floor swabbing. No positive floor samples for SARS-CoV-2 were detected (Fig 1B). Air filter sampling of single HEPA units in all rooms except Rooms 2, 4, and 7 began on May 20 and continued until August 12, 2021 with no positive samples detected.

In School C (n = 130 swabs), eleven rooms were monitored weekly from March 2 to May 25, 2021 through floor swabbing. Preliminary sampling was conducted on February 23, 2021 when diverse surfaces were sampled in the same eleven rooms, and all of these samples tested negative. The only positive sample for SARS-CoV-2 throughout the weekly sampling was collected in Room 2 on April 20, 2021 (Fig 1B). Unlike results in School A, no repeated positives in the same room were observed. All teachers, staff and students associated with the positive room were tested for COVID-19 after the positive floor swab, but no clinical cases were found. A family member of a Room 2 occupant did test positive following the environmental detection (personal communication with school administrator). This positive test for a family member raises an important possibility that needs to be considered: SARS-CoV-2 RNA could be shed by an "outsider" (i.e., not someone in the school) and then brought into the sampled environment by someone who is not actively shedding. The mechanical transfer of SARS-CoV-2 RNA on fomites has not been rigorously tested; however, viral RNA has been detected on personal items like clothes, towels, bedding, mobile phones, and shoe soles [36–39]. Transfer of outsider SARS-CoV-2 genetic material to sampled rooms is thus an important possibility that should be considered in the context of environmental monitoring.

### 3.3 Floor and HEPA filter swab sampling for the detection of SARS-CoV-2

We established a different environmental monitoring strategy in Schools D (n = 680 swabs) and E (n = 763 swabs), informed by the results from the HEPA filter sampling after a known exposure test detailed in section 3.1. From the first day of sampling, we collected one floor and two HEPA filter samples (one sample from each of two HEPA filtration units) per room. In School D, preliminary testing episodes were conducted in Rooms 11 and 20 on March 10, 16 and 30, 2021. All HEPA filter and floor samples collected during these episodes were negative for SARS-CoV-2. The weekly sampling in School D covered 25 rooms and ran from April 13 to June 8, 2021. Ten HEPA filter samples and one floor sample collected during this period tested positive (Fig 2A). All positive detections were from a single filter or floor sample at a time; that is, no more than one sample per room ever tested positive for SARS-CoV-2 during the same sampling episode. To the best of our knowledge, this is the first study to report portable HEPA filter sampling as a strategy for detecting environmental SARS-CoV-2 RNA. Air sampling through the use of air filter samplers, electrostatic precipitators and HVAC filters has been demonstrated [15, 31, 32, 40]. Positive air and surface samples in schools have also been previously reported [40, 41]. Limited participation in on-site clinical testing was observed in School D after the reporting of positive environmental results on most episodes; therefore, no direct links were established between environmental and clinical results. Participation in testing varied between schools and no data is available for schools A, B, and C. Testing in School D varied week to week from 33% to 88% with similar but slightly lower testing rates at School E.

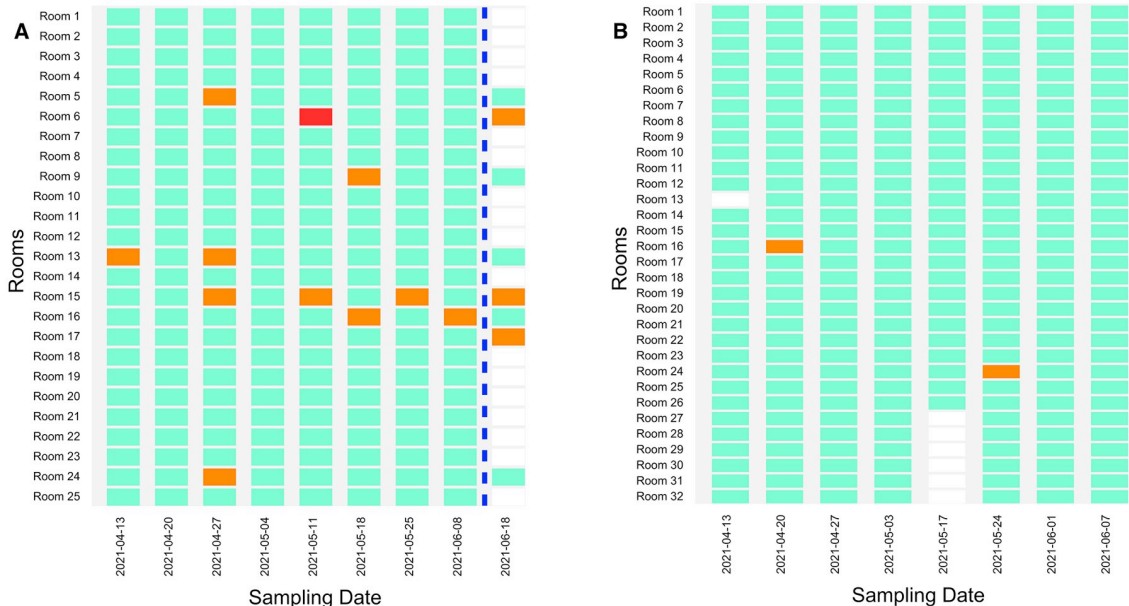

**Fig 2.** Positive and negative rooms for SARS-CoV-2 based on floor and HEPA filter samples collected in (**A**) School D, and (**B**) School E throughout the environmental monitoring study. Testing dates with a positive floor sample are in red, episodes with a positive air filter sample are in orange, negative episodes are in green, and episodes where no samples were collected are in white. No more than one sample tested positive in a room at any given time. Six negative samples collected from Rooms 27–32 in School E on May 17, 2021 were impossible to link to specific rooms because the tube labels were compromised (not included in the figure). A concluding sampling episode was conducted in School D on June 18, 2021 after school sessions ended to gather further information on the persistence of environmental SARS-CoV-2 RNA in previously positive classrooms and is indicated by the blue line. The previously negative Room 17 was included in the concluding sampling episode as a negative control.

A concluding sampling episode was conducted in School D on June 18, 2021, eight days after the last day of the school year. The goal of this sampling episode was to gather further information on the persistence of environmental SARS-CoV-2 RNA in previously positive classrooms. Rooms 5, 6, 9, 13, 15, 16, 17 and 24 were sampled. Each of these rooms except for Room 17 (selected as a negative control) yielded a positive detection of SARS-CoV-2 at some point during the weekly sampling. During the concluding sampling episode, samples from a door jamb, teacher desk and center surface (either a desk, chair, podium, or projector located in the center of the classroom) were collected from each room alongside the two filter samples and single floor sample. One filter sample from Room 6 and one filter sample from Room 15 tested positive once more (Fig 2A). Unexpectedly, a filter sample from Room 17 (the negative control) tested positive as well. These results further confirmed the challenges of interpreting positive results in previously positive environments, since positive results could be caused by the resuspension and capture of relic RNA on air filters.

In School D, we replaced HEPA filters after a positive detection to mitigate the impact of relic RNA contamination on future samples and avoid the issue of repeated positives observed in School A. The strategy was not successful as repeated SARS-CoV-2 positives were observed in three rooms after new filters were installed. Rooms 6, 13, 15, and 16 yielded two or four repeated intermittent positives during the study, including the concluding sampling episode (Fig 2A). The longest period of intermittent repeated positives was in Room 15, covering 52 days. Even with the change in filters after a positive environmental detection, capturing relic RNA through HEPA filter sampling remained a possibility. The intermittent and repeated positives could have been a result of the resuspension of dust particles containing SARS-CoV-2

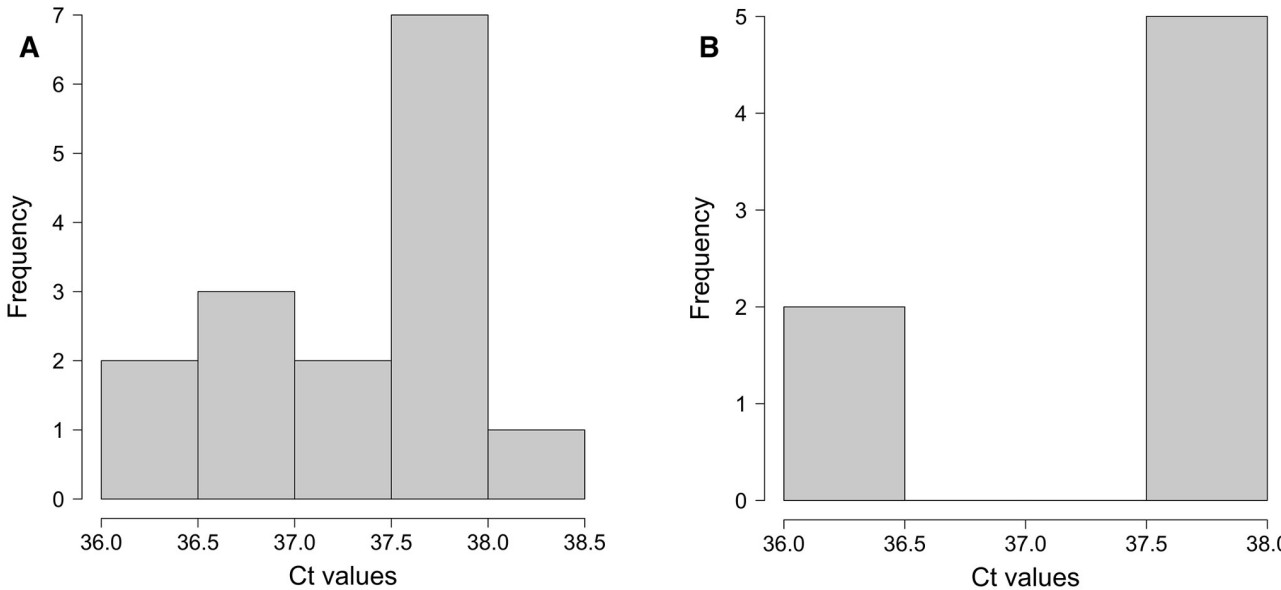

**Fig 3.** Frequency distributions of cycle threshold (Ct) values for positive samples from (**A**) HEPA filters and (**B**) floors at Schools A, B, C, and D. All positive samples were processed using the MagMAX automated extraction protocol and tested using a SARS-CoV-2 S gene RT-qPCR assay.

genetic material. SARS-CoV-2 RNA has been detected in floor and HVAC dust up to two months after patient symptom onset [15], which could explain our results in the absence of clinical confirmation due to the low testing participation.

In School E, we conducted preliminary testing episodes on March 10, 16, and 30, 2021 in a room that was not part of the weekly sampling campaign. The air filter and floor samples collected during the preliminary testing were negative for SARS-CoV-2. Weekly sampling covering 32 rooms was conducted from April 13 to June 7, 2021. As in School D, we collected two HEPA filter samples and one floor sample from each School E classroom during each sampling episode. A HEPA filter replacement strategy was also implemented. Of all the samples collected during the campaign, only two—a single filter sample each from Rooms 16 and 24—tested positive for SARS-CoV-2 (Fig 2B). No repeated positives were observed. Site-specific conditions may contribute to the resuspension of dust particles, including room ventilation [42], which could explain the fact that intermittent positives were observed in some classrooms but not others. Increased on-site clinical testing participation occurred in School E after the positive environmental results were reported, but no positive individuals were identified.

### 3.4 High Ct values in floor and air filter swab samples

qPCR Ct values are inversely proportional to the concentrations of the target genes in the samples tested. Observed Ct values across all filter and floor samples collected for this study ranged from 36.07–38.01 (Fig 3A and 3B), which is close to a commonly used threshold for considering a sample positive (Ct<40) [4, 30]. The high Cts obtained were likely due to low amounts of virus collected through the environmental sampling methods. It is also possible that all the positives detected in the present study were from relic genetic material, as no infected individuals were found through clinical testing. It is worth noting that infectious virus has not been recovered from environmental samples with Ct values above 30 [14, 43].

The Ct values measured in this study are in the range of what has been observed for SARS-CoV-2 surface sampling studies. Ct values >30 were reported in a clinical environment with

COVID-19 patient and non-patient care areas [14], and a median Ct of 35 and interquartile range of 34–36.5 were reported in quarantine environments [35] 34–44 in schools [41], 34–38 in workplace sites [5] and 29.0–38.1 in public locations such as public squares and bus terminals [29]. Similarly, Ct values in the 36 to 39 range were reported from air samples collected with glass fiber filters in a clinical setting [31]. In the present study, air filter Ct values (Ct 36.37–38.01) were similar in magnitude but more evenly distributed across the detection range than floor samples (Ct 36.07–37.85) (Fig 3). The similar Ct results in air filter and floor samples suggest comparable amounts of virus deposited on the two surface types.

### 3.5 Surface and air filter sampling as strategies to inform the management of the COVID-19 pandemic

Even though no clinical positive cases of COVID-19 were identified following the detection of environmental positives in Schools D and E, clear differences were observed in the environmental positivity rates of the two schools. For School D, the mean environmental positivity rate was 5.5% positive rooms per sampling episode (considering filter and floor samples together, and excluding the concluding sampling session). This positivity rate is seven times higher than the 0.78% positivity rate observed for School E. Self-reported positive case data from Schools D and E for the 2021 school year was also provided to us by the school district. No positives were reported during the environmental sampling period (April 13 to June 8, 20201) in either of the two schools. However, two positive cases were reported in School D on February 1 and 3, 2021 during limited in-person instruction. No cases were reported during in-person instruction in School E. Relationships between environmental surface positivity rates and clinical cases in larger populations have been established [4]. The identification of prior positive cases in School D highlights the potential link between clinical and environmental results, while also accentuating the potential for relic RNA detection in environmental samples during periods of low case positivity rates.

The two strategies utilized in this study for SARS-CoV-2 monitoring in the environment—floor and HEPA filter sampling—also yielded different positivity rates. There were 15 identifiable air filter positives throughout the sampling period in Schools D and E (including the final sampling episode in School D) but only a single positive floor sample. Two HEPA filter samples were collected from each room, compared to one floor swab. The HEPA filter positivity rates in Schools D and E were 3.02% and 0.41%, respectively, while the floor positivity rates were 0.48% and 0%. These results suggest that HEPA filter sampling is more sensitive than floor sampling. Air filters collect and concentrate particles that can contain SARS-CoV-2—including aerosols, droplets, and dust—on a relatively small surface area. While such particles are also deposited on floors, a smaller fraction of the floor surface area was sampled, and floors were frequently cleaned by school staff.

There are clear challenges with using surface and filter sampling to monitor the presence of SARS-CoV-2. Many of these challenges, including near-limit-of-detection virus concentrations and the impacts of site-specific conditions, have also been reported for other types of environmental monitoring, such as wastewater surveillance [44]. The most significant challenge we encountered was the differentiation between freshly shed, relic, or outsider SARS-CoV-2. Clinical testing as a response to positive environmental RNA can be useful to identify active shedders of the virus. Yet all individuals in the space may still test negative when environmental signals result from relic or outsider RNA. If full clinical testing participation is not possible nor available, SARS-CoV-2 environmental surface monitoring may offer useful public health information by establishing background environmental detections and tracking

changes in positive rates through time, but cannot differentiate between freshly shed, relic, or outsider RNA.

## 4 Conclusions

Portable HEPA filter and floor sampling are environmental monitoring tools that can successfully detect SARS-CoV-2 RNA. In this study we collected 2,341 swabs from both surfaces and HEPA filters, detecting positive samples in 20 classroom sampling events. HEPA filter sampling was a more sensitive technique compared to collection of floor swabs. In schools or other settings where access to or participation in clinical testing programs is limited, HEPA filter testing could be a useful strategy to inform pandemic response. However, environmental monitoring of SARS-CoV-2 through surface sampling (including HEPA filters) poses the challenge of differentiating amongst fresh, relic, and outsider viral RNA, especially for high-Ct results. Further research is needed to establish Ct thresholds for HEPA filter monitoring that indicate nearby active infections and elevated exposure risks. New technologies and testing protocols that differentiate fresh from aged viral RNA would also do much to increase the utility of SARS-CoV-2 surface monitoring in schools.

## Supporting information

**S1 File. Supplemental information (1–4) describes the validation, instructional videos, extraction protocols and standard curves.** S1 Table shows the RT-qPCR standard curve and S2 Table lists the samples used in validation.
(DOCX)

## Acknowledgments

The authors would like to thank the many teachers, administrators, staff, parents, and others associated with these schools for facilitation of sample collection. We also thank The Biology and Built Environment Center at the University of Oregon for supplying expertise and reagents at the start of this project. Members of the Bischel lab provided valuable laboratory support and feedback on the manuscript.

## Author Contributions

**Conceptualization:** Rogelio Zuniga-Montanez, David A. Coil, Jonathan A. Eisen, Minji Kim, Karen Shapiro, Heather N. Bischel.

**Data curation:** Rogelio Zuniga-Montanez, David A. Coil, Randi Pechacek.

**Formal analysis:** Rogelio Zuniga-Montanez, David A. Coil, Randi Pechacek, Minji Kim, Heather N. Bischel.

**Funding acquisition:** David A. Coil, Heather N. Bischel.

**Investigation:** Rogelio Zuniga-Montanez, David A. Coil, Randi Pechacek, Roque G. Guerrero, Minji Kim, Heather N. Bischel.

**Methodology:** Rogelio Zuniga-Montanez, David A. Coil, Jonathan A. Eisen, Randi Pechacek, Roque G. Guerrero, Minji Kim, Heather N. Bischel.

**Project administration:** Rogelio Zuniga-Montanez, David A. Coil, Karen Shapiro, Heather N. Bischel.

**Supervision:** Rogelio Zuniga-Montanez, David A. Coil, Jonathan A. Eisen, Karen Shapiro, Heather N. Bischel.

**Validation:** Rogelio Zuniga-Montanez, Randi Pechacek, Minji Kim.

**Visualization:** Rogelio Zuniga-Montanez.

**Writing – original draft:** Rogelio Zuniga-Montanez, David A. Coil, Randi Pechacek, Roque G. Guerrero.

**Writing – review & editing:** Rogelio Zuniga-Montanez, David A. Coil, Jonathan A. Eisen, Randi Pechacek, Roque G. Guerrero, Minji Kim, Karen Shapiro, Heather N. Bischel.

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
