## [Decision Letter · Decision Letter 0]

17 Jan 2022

PONE-D-21-35360The challenge of SARS-CoV-2 environmental monitoring in schools using floors and portable HEPA filtration units: Fresh or relic RNA?PLOS ONE

Dear Dr. Coil,

Thank you for submitting your manuscript to PLOS ONE. After careful consideration, we feel that it has merit but does not fully meet PLOS ONE’s publication criteria as it currently stands. Therefore, we invite you to submit a revised version of the manuscript that addresses the points raised during the review process.

 The reviewers raised important considerations that need to be addressed. Please respond to each comment and revised your manuscript accordingly.

We look forward to receiving your revised manuscript.

Kind regards,

Luisa Gregori, Ph.D

Academic Editor

PLOS ONE

Journal Requirements:

a) Did participants provide their written or verbal informed consent to participate in this study?

b) If consent was verbal, please explain i) why written consent was not obtained, ii) how you documented participant consent, and iii) whether the ethics committees/IRB approved this consent procedure."

“Funding was provided by Healthy Yolo Together/Healthy Davis Together, a partnership between the University of California, Davis, the city of Davis, and Yolo county.”

“Funding was provided by Healthy Yolo Together/Healthy Davis Together. The funders had no role in study design, data collection and analysis, decision to publish, or preparation of the manuscript.”

Reviewers' comments:

Reviewer's Responses to Questions

**Comments to the Author**

1. Is the manuscript technically sound, and do the data support the conclusions?

Reviewer #1: Yes

Reviewer #2: Yes

2. Has the statistical analysis been performed appropriately and rigorously? 

Reviewer #1: N/A

Reviewer #2: Yes

3. Have the authors made all data underlying the findings in their manuscript fully available?

Reviewer #1: No

Reviewer #2: Yes

4. Is the manuscript presented in an intelligible fashion and written in standard English?

Reviewer #1: Yes

Reviewer #2: Yes

5. Review Comments to the Author

Reviewer #1: This is a very interesting manuscript, but it is necessary to focus on a few problematic passages.

The methodology, not the supplementary materials, should indicate which portable air purifier was used. For used air purifier should be verified the air flow in cubic meters per hour and should be indicated how long the air has been sampled. The authors should calculate the approximate area of the sampled part of the HEPA filter in square centimeters.

Why did the authors used the Spike protein RNA detection method? It is the most mutating part of SARS-CoV-2 coronavirus and the accuracy and sensitivity of some methods using spike protein detection is low. Can the authors analyze the primers and probes using the GISAID database and determine if the primers and probes are designed for regions that are not mutated? This information should be provided in Methods.

Could the authors create a calibration line for RealTime PCR according to the standard and realistically calculate the number of detected viruses (RNAs) in each PCR positive reaction? A table with the numbers of viruses in each sample should be provided, not Ct values. It is not possible to compare Ct values between different methodologies. The introductory paragraph of Chapter 3.1 is a bit confusing, as each methodology has a different Ct value for capturing the same amount of DNA. Therefore, Ct values cannot be compared. This paragraph should be redrafted in this respect.

If the analysis of some rooms showed positivity alternating with negativity, how did the authors rule out PCR contamination in the laboratory or at collection? It should be described in manuscript.

Is it really a reusable air purifier that captured coronavirus RNA even after changing the filter? How did authors decontaminate from the RNA mesh and cover where RNAs can also be captured?

How is the cleaning done in the classrooms – every day? By which method? Is the disinfectant used for floor cleaning and if so, which one? It is possible that the same mop is used in toilets and this mop can then spread viral RNA in other rooms in the school. For example, one of the school staff could be positive and spread viruses in the toilets.

The whole section 3. Results and discussion I recommend to rewrite to two chapters: the results chapter of the discussion chapter. Unfortunately, this chapter turned out to be confusing for readers. The reader will not know whether these are the results of this work or a citation of other results. The separate chapter Results will greatly contribute to the clarity of the results.

The abstract cannot state: "The high-Ct positive results from environmental swabs further suggest the absence of active infections", if the authors did not determine the viability of the virus. The abstract should indicate the number of viruses, not the Ct values.

If the abstract compares air sampling and square centimeters on the floor, for the air sampling should indicate the air flow (cubic meters per hour) through the HEPA filter and samples surface of the HEPA filter.

The methodology lacks an indication how children and teachers were tested for positivity. Missing is used RealTime PCR method and the sampling method. Saliva, oropharyngeal swabs, nasopharyngeal swabs?

Reviewer #2: This work is an important research methodology that shows the possibility of SARS-CoV-2 surface detection (surveillance) from floors and portable filters. I enjoyed reading the manuscript, and I have a few minor comments below that I believe will help improve the manuscript. I accept this manuscript for publication after addressing my comments below.

Introduction

1. First paragraph. Introduce COVID-19 in one sentence.

2. First paragraph, last sentence, add a reference.

3. Paragraph 3: Wastewater surveillance cannot provide information about virus presence in interior spaces (e.g., location inside the building floors and rooms). That is correct. However, you did not mention the latest study by Sousan et al. 2021 (below) that suggests sampling HVAC systems as a surveillance method. You mention the HVAC system in the next paragraph. How does this work address the limitation(s) or difference(s) in Sousan et al. (2021)?

Sousan, S., Fan, M., Outlaw, K., Williams, S., & Roper, R. L. (2021). SARS-CoV-2 Detection in Air Samples from Inside Heating, Ventilation, and Air Conditioning (HVAC) Systems-COVID Surveillance in Student Dorms. American journal of infection control.

Materials and Methods

4. Section 2.2: I suggest removing the words “Pilot study.” Five schools from January to August is a lot of work.

Results

5. Before you start the sections, it would be nice to mention the total number of samples collected and maybe a breakdown of each method and location by sample. Before you mention the details, this will show the magnitude of your work.

6. Section 3.1 (first paragraph): The Methods section should mention the ten surface swab samples.

7. Section 3.1 (first paragraph): I am confused about this sentence “First, we sampled surfaces at a house where an asymptomatic person who later tested positive for COVID-19 was present for 2 hours.” In the previous sentence, you mention “at least one COVID-19 positive individual was known”. Why did you sample in that house if you did not know the person had COVID-19 (asymptomatic)? Sampling in a school makes sense because of the students’ exposure between their home and the classroom. Therefore, the likelihood of detecting COVID-19 is high. Was sampling in this house randomly, and how did you choose this random house? Also, in the Methods section, you mention “using opportunistic sampling in two locations within six days after one or more clinical COVID-19 cases were identified”. So in the Methods, you mention that sampling started after identifying COVID-19, and here you say that you sampled in the house and then later you found out that the person had covid-19. Maybe there is a wording issue; please re-write for clarity. Please also address this comment in the Methods section.

8. Section 3.1 (second paragraph): Did you swab underneath the chairs for all students randomly or just the ones that tested positive?

9. Section 3.1 (third paragraph): give an example using the Sousan et al. (2021) study (above) to validate your point.

10. Section 3.2 (first paragraph: last four sentences): Is this a false-positive result, and is this considered a limitation of the surveillance method? Please clarify.

11. Figures 1 and 2 are blurry and sufficient for publication. Please improve the quality of the figure considerably and resubmit to the journal.

12. In the end, how does this work compare to other SARS-CoV-2 surveillance methods? Please discuss?

Conclusion

13. Please add the number of positive samples (your PCR results) and the number of confirmed positive samples (clinical PCR testing) for your positives.

6. PLOS authors have the option to publish the peer review history of their article (what does this mean?). If published, this will include your full peer review and any attached files.

Reviewer #1: **Yes: **Omar Šerý

Reviewer #2: No

---

## [Author Response · Author response to Decision Letter 0]

14 Mar 2022

The response to reviewers has been attached to this re-submission.

---

## [Decision Letter · Decision Letter 1]

5 Apr 2022

The challenge of SARS-CoV-2 environmental monitoring in schools using floors and portable HEPA filtration units: Fresh or relic RNA?

PONE-D-21-35360R1

Dear Dr. Coil,

We’re pleased to inform you that your manuscript has been judged scientifically suitable for publication and will be formally accepted for publication once it meets all outstanding technical requirements.

Kind regards,

Luisa Gregori

Academic Editor

PLOS ONE

Additional Editor Comments (optional):

Reviewers' comments:

Reviewer's Responses to Questions

**Comments to the Author**

1. If the authors have adequately addressed your comments raised in a previous round of review and you feel that this manuscript is now acceptable for publication, you may indicate that here to bypass the “Comments to the Author” section, enter your conflict of interest statement in the “Confidential to Editor” section, and submit your "Accept" recommendation.

Reviewer #1: All comments have been addressed

Reviewer #2: All comments have been addressed

2. Is the manuscript technically sound, and do the data support the conclusions?

Reviewer #1: Yes

Reviewer #2: Yes

3. Has the statistical analysis been performed appropriately and rigorously? 

Reviewer #1: N/A

Reviewer #2: Yes

4. Have the authors made all data underlying the findings in their manuscript fully available?

Reviewer #1: Yes

Reviewer #2: No

5. Is the manuscript presented in an intelligible fashion and written in standard English?

Reviewer #1: Yes

Reviewer #2: Yes

6. Review Comments to the Author

Reviewer #1: The authors of the manuscript answered all the questions and incorporated into the manuscript the changes that were needed.

Reviewer #2: (No Response)

7. PLOS authors have the option to publish the peer review history of their article (what does this mean?). If published, this will include your full peer review and any attached files.

Reviewer #1: **Yes: **Omar Šerý

Reviewer #2: No

---

## [Editor Report · Acceptance letter]

13 Apr 2022

PONE-D-21-35360R1 

The challenge of SARS-CoV-2 environmental monitoring in schools using floors and portable HEPA filtration units: Fresh or relic RNA? 

Dear Dr. Coil:

I'm pleased to inform you that your manuscript has been deemed suitable for publication in PLOS ONE. Congratulations! Your manuscript is now with our production department. 

Kind regards, 

on behalf of

Dr Luisa Gregori 

Academic Editor

PLOS ONE